# Nationwide survey on Japanese residents' experience with and barriers to incident reporting

Masaru Kurihara[1], Takashi Watari[2,3,4]*, Jeffrey M. Rohde[3,4], Ashwin Gupta[3,4], Yasuharu Tokuda[5], Yoshimasa Nagao[1]

1 Department of Patient Safety, Nagoya University Hospital, Nagoya, Japan, 2 General Medicine Center, Shimane University Hospital, Izumo, Shimane, Japan, 3 Medicine Service, VA Ann Arbor Healthcare System, Ann Arbor, Michigan, United States of America, 4 Department of Medicine, University of Michigan Medical School, Ann Arbor, Michigan, United States of America, 5 Tokyo Foundation for Policy Research, Tokyo, Japan

* wataritari@gmail.com

**Data Availability Statement:** The data that support the findings of this study are available from the General Medicine Center, Shimane University Hospital (E-mail. shimanegp@gmail.com), upon

## Abstract

The ability of any incident reporting system to improve patient care is dependent upon robust reporting practices. However, under-reporting is still a problem worldwide. We aimed to reveal the barriers experienced while reporting an incident through a nationwide survey in Japan. We conducted a cross-sectional survey. All first- and second-year residents who took the General Medicine In-Training Examination (GM-ITE) from February to March 2021 in Japan were selected for the study. The voluntary questionnaire asked participants regarding the number of safety incidents encountered and reported within the previous year and the barriers to reporting incidents. Demographics were obtained from the GM-ITE. The answers of respondents who indicated they had never previously reported an incident (non-reporting group) were compared to those of respondents who had reported at least one incident in the previous year (reporting group). Of 5810 respondents, the vast majority indicated they had encountered at least one safety incident in the past year (n = 4449, 76.5%). However, only 2724 (46.9%) had submitted an incident report. Under-reporting (more safety incidents compared to the number of reports) was evident in 1523 (26.2%) respondents. The most frequently mentioned barrier to reporting an incident was the time required to file the report (n = 2622, 45.1%). The barriers to incident reporting were significantly different between resident physicians who had previously reported and those who had never previously reported an incident. Our study revealed that resident physicians in Japan commonly encounter patient safety incidents but under-report them. Numerous perceived and experienced barriers to reporting remain, which should be addressed if incident reporting systems are to have an optimal impact on improving patient safety. Incident reporting is essential for improving patient safety in an institution, and this study recommends establishing appropriate interventions according to each learner's barriers for reporting.

reasonable request. This is because that the data contain potentially identifying.

**Funding:** This work was supported by the national academic research grant funds [JSPS KA-KENHI: 17K15745, 20H03913]. The sponsor of the study had no role in the study design, data collection, analysis, or preparation of the manuscript.

**Competing interests:** The authors have no conflicts of interest to declare. All authors have reviewed and agree with the contents of the manuscript, and there are no financial interests to report.

## Introduction

Since the launch of the modern patient safety movement more than two decades ago, marked by the publication of the Institute of Medicine's "To Err is Human," healthcare systems have undertaken a variety of initiatives with the goal of making healthcare safer [1]. Among the most pervasive efforts, incident reporting systems aspire to identify and record adverse events or near misses, facilitate learning, and enable the implementation of countermeasures to prevent recurrences [2]. Beyond individual systems, some nations, such as the United Kingdom and Japan, aggregate data from all incident reporting systems to develop interventions aimed at preventing recurrences nationwide [3, 4]. For this instrument of change to be effective, a robust safety culture must be fostered, such that front-line healthcare workers report when they see something happening.

Resident physicians are uniquely situated among front-line healthcare workers given their variety of practice settings and frequent interactions with patients and families, often across institutions. Therefore, they are often the first to encounter safety incidents [5–7]. Unfortunately, incident reporting rates among physicians, including residents, is low, as shown in multiple studies, representing fewer than 5% of reports [8, 9]. Additional training highlighting the process and benefits of incident reporting is important. It has even been a basic requirement for the completion of residency training in Japan [10]. However, such educational interventions alone have been insufficient in significantly impacting patient safety practices [11–14]. In fact, despite these efforts, a recent survey showed that half of the resident physicians had not submitted an incident report in the past year, and more than half did not even know how to submit an incident report [13].

To gain a broader understanding of reporting patterns and barriers experienced while reporting, we conducted a nationwide survey of residents in Japan. Particularly, we compared the perceived barriers to reporting for residents who had recently reported incidents compared to non-reporters. Developing countermeasures aimed at perceived barriers felt by non-reporters could help broaden resident physician engagement in patient safety. This could also address the barriers experienced by previous reporters and help optimize the system and encourage subsequent reporting.

## Methods

### Study design

This study was a nationwide, cross-sectional survey in Japan. Based on a previous study [13], we used a validated questionnaire on patient safety, which is to be completed at the end of the General Medicine In-Training Examination (GM-ITE). The GM-ITE, designed by a committee of the Japan Institute for Advancement of Medical Education Program (JAMEP), provides program directors with an objective and reliable assessment of a resident's fundamental clinical knowledge. After the GM-ITE, the participants completed an optional questionnaire that assessed their residency training and work environment, including their incident reporting behavior. Both the original GM-ITE and the abovementioned questionnaire have been used in prior studies [13, 15, 16]. This study was approved by the Ethics committee of Japan Institute for Advancement of Medical Education Programme (20–2). Informed consent was obtained from all participants in the written form.

### Study participants

The study included 7669 residents who worked in 593 medical institutions nationwide and took the GM-ITE in February and March 2021. In 2004, a new residency system was enacted,

under which Japanese law requires all physicians to spend two years in residency. Based on this system, physicians with post-graduate years (PGY) 1 and 2 are called residents in Japan. Those who did not agree to participate in the survey or with missing data from the clinical training environment survey questionnaire or examinee characteristics were excluded from the analysis.

## Data collection

The questionnaire was developed following consensus among two investigators based on known incident reporting challenges in Japan (e.g., under-reporting and lack of knowledge on patient safety) [13]. First, the questionnaire asked participants about the total number of incidents they had encountered and the number of incidents they had reported in the previous year. A patient safety incident was defined as "any unintended or unexpected incident that could have led, or did lead, to harm for one or more patients receiving healthcare" [17]. Moreover, we added questions regarding the barriers to incident reporting based on a previous report [18]. The items of this questionnaire were classified into eight parts as follows:

a. It takes time to report.

b. Even if I report, no improvement will be made anyway.

c. I do not know the criteria for reporting.

d. I do not know the reporting procedure.

e. I do not get any feedback even if I report.

f. I feel that I will be punished if I report.

g. I feel mentally burdened when I report.

h. Because senior doctors tend not to report.

In addition to the patient safety questionnaire, residents' demographic data (e.g., age, PGY [1 or 2], and hospital) were collected. Hospital information (hospital type [university or community-based] and location) was obtained from the Japan Residency Matching Program website [19] and the Foundation for the Promotion of Medical Training website [20]. Regarding the categories of hospital locations, 20 cities designated by the Ministry of Internal Affairs and Communications and the 23 wards in Tokyo were defined as urban cities, while the rest were defined as provincial cities.

## Statistical analyses

Program directors of each hospital collected the GM-ITE answer sheets and questionnaire survey form after the exam was completed and returned them in the provided secured envelope. Data were collected and anonymized from the web database by an independent data manager. Subsequently, responses regarding patient safety activities between residents who never experienced incident reporting (non-reporting group) and those who experienced incident reporting at least once in the previous year (reporting group) were compared. Intergroup differences in statistical data were assessed using Mann-Whitney U tests and chi-square tests for continuous and categorical variables, respectively. Statistical analysis was performed using STATA version 11 (Stata Corporation, College Station, TX, USA), and statistical significance was defined at $P < 0.05$.

## Results

### Characteristics of respondents and hospitals

A total of 7669 initial residents from 593 hospitals participated in the GM-ITE. Of these, 853 residents who did not agree to participate and 1006 residents with missing data were excluded, yielding 5810 respondents. Fig 1 presents the respondent flow diagram.

Table 1 summarizes the characteristics of the respondents and hospitals.

### Incident reporting during residency

Of the 5810 respondents, 3086 residents (53.1%) reported that they had not submitted an incident report over the previous one year. There were 1448 (24.9%) and 547 respondents (9.4%) who reported that they had submitted one or two incident reports over the previous year, respectively (Table 2).

### Encountering incidents during residency

A total of 1361 respondents (23.4%) reported that they had not encountered any safety incidents over the previous year. There were 1907 respondents (32.8%) who reported that they

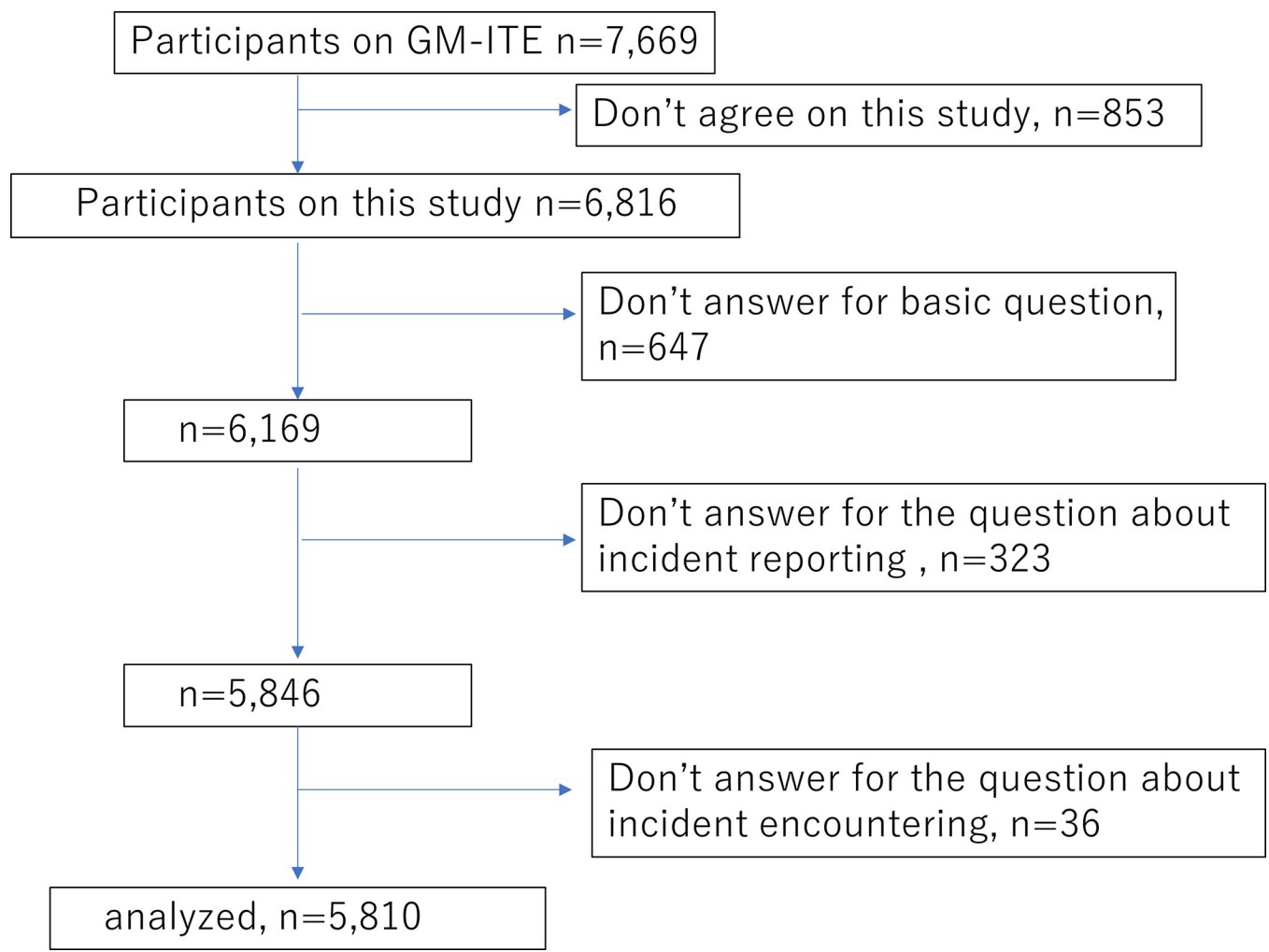

**Fig 1. Flowchart of survey participants.**

**Table 1. Respondent characteristics by experience of incident reporting.**

| | Incident reporting per year | | | | | | |
|---|---|---|---|---|---|---|---|
| | **No Report** | **1 time** | **2 times** | **3 times** | **4 times** | **> 5 times** | **Total n(%)** |
| **Resident characteristics** | | | | | | | |
| **Sex** | | | | | | | |
| Male | 2,090 | 989 | 365 | 160 | 65 | 286 | 3,995(68.1) |
| Female | 996 | 459 | 182 | 59 | 32 | 127 | 1,855(31.9) |
| **PGY** | | | | | | | |
| PGY1 | 1,593 | 659 | 286 | 131 | 60 | 226 | 2,955(50.9) |
| PGY2 | 1,493 | 789 | 261 | 88 | 37 | 187 | 2,855(49.1) |
| **Hospital characteristics** | | | | | | | |
| **Hospital Location** | | | | | | | |
| Urban | 2,102 | 944 | 351 | 143 | 64 | 280 | 3,884(66.9) |
| Rural | 984 | 504 | 196 | 76 | 33 | 133 | 1,926(33.1) |
| **Hospital Type** | | | | | | | |
| Community-based hospital | 2,720 | 1,290 | 491 | 188 | 86 | 370 | 5,145(88.6) |
| University hospital | 366 | 158 | 56 | 31 | 11 | 43 | 665(11.4) |

*Note*: PGY: post-graduate year

had encountered one incident over the previous year. Table 2 shows the relationship between the experiences of incident reporting and encountering an incident in a year.

## Barriers to incident reporting

Among all respondents, the most frequently reported barrier to incident report completion was the time required to file the report (n = 2622, 45.1%), followed by lack of knowledge on the criteria for incident reporting (n = 1888, 32.5%) (Table 3).

There were 373 residents (6.3%) who responded that they feared punishment if they reported an incident. While comparing the non-reporting and reporting groups, those who did not report were more likely to cite lack of knowledge on reporting criteria and procedure, fear of punishment, mental burden associated with reporting, and lack of example by senior

**Table 2. Relationship between the experiences of incident reporting and encountering incidents in a year.**

| | | Experiences of incident reporting in a year | | | | | | |
|---|---|---|---|---|---|---|---|---|
| **Encountering safety incidents in a year** | | None | 1 | 2 | 3 | 4 | 5 or more | Total (%) |
| None | | 1,244 | 86 | 19 | 2 | 1 | 9 | 1,361 |
| | | 91.4 | 6.32 | 1.4 | 0.15 | 0.07 | 0.66 | 100 |
| 1 | | 1,010 | 760 | 72 | 17 | 12 | 36 | 1,907 |
| | | 52.96 | 39.85 | 3.78 | 0.89 | 0.63 | 1.89 | 100 |
| 2 | | 559 | 371 | 269 | 29 | 15 | 40 | 1,283 |
| | | 43.57 | 28.92 | 20.97 | 2.26 | 1.17 | 3.12 | 100 |
| 3 | | 140 | 116 | 99 | 93 | 12 | 29 | 489 |
| | | 28.63 | 23.72 | 20.25 | 19.02 | 2.45 | 5.93 | 100 |
| 4 | | 26 | 20 | 18 | 20 | 34 | 11 | 129 |
| | | 20.16 | 15.5 | 13.95 | 15.5 | 26.36 | 8.53 | 100 |
| 5 or more | | 107 | 95 | 70 | 58 | 23 | 288 | 641 |
| | | 16.69 | 14.82 | 10.92 | 9.05 | 3.59 | 44.93 | 100 |

**Table 3. Barriers to incident reporting.**

| | Total (%) | Incident reporting experience | | | | |
|---|---|---|---|---|---|---|
| | | no | % | Yes | % | p-value |
| Q1: It takes time to report. | 2,622(45.1) | 1,208 | 39.14 | 1,414 | 51.91 | <0.001 |
| Q2: Even if I report, no improvement will be made anyway. | 386(6.6) | 160 | 5.18 | 226 | 8.3 | <0.001 |
| Q3: I do not know the criteria for reporting. | 1,888(32.5) | 1,212 | 39.27 | 676 | 24.8 | <0.001 |
| Q4: I do not know the reporting procedure. | 1,044(18.0) | 892 | 28.90 | 152 | 5.58 | <0.001 |
| Q5: I do not get any feedback even if I report. | 571(9.8) | 228 | 7.39 | 343 | 12.59 | <0.001 |
| Q6: I feel that I will be punished if I report. | 363(6.3) | 212 | 6.87 | 151 | 5.54 | 0.04 |
| Q7: I feel mentally burdened when I report. | 778(13.4) | 459 | 14.87 | 459 | 11.71 | <0.001 |
| Q8: Because senior doctors tend not to report. | 509(8.8) | 307 | 9.95 | 202 | 7.42 | 0.001 |

physicians as barriers to reporting. Those with experience reporting more often cited time required to report and lack of improvement or feedback after a report as barriers to reporting (Table 3). Fig 2 shows the differences in the barriers depending on reporting experience.

## Discussion

In this nationwide survey in Japan, more than three-quarters of the resident physicians indicated that they had experienced at least one safety incident in the past year. However, less than half had filed an incident report in that time. Numerous barriers were noted, as shown in Fig 2, with significant differences between the non-reporting and reporting groups. Non-reporting residents indicated that they were less familiar with the details of the reporting system, had not seen senior physicians report incidents, and more commonly felt mentally burdened with reporting safety incidents. Residents who had previously reported safety incidents more commonly mentioned the time burden and the lack of feedback and noticeable improvement as a result of the reporting as barriers. These findings reveal that under-reporting of patient safety incidents continues to be a widespread issue among general medicine resident physicians due to both perceived and experienced barriers, and it should be addressed urgently to optimize this process.

Further evidence of under-reporting suggests that more than a quarter of residents indicated that the number of safety incidents they had encountered in the past year was greater

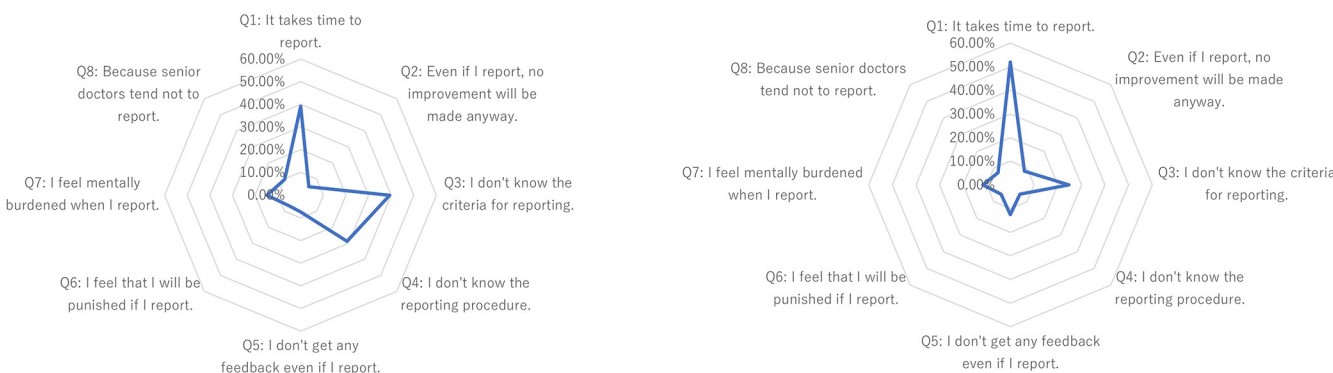

**Fig 2. Radar chart of barriers to incident reporting.** Left: Participants who never experienced incident reporting. Right: Participants who experienced incident reporting at least once a year.

than the number of reports they had filed (by more than 4000 total events). This under-reporting of incidents by physicians at all levels of training has been recognized as a longstanding and pervasive problem. In a survey of internal medicine house staff and faculty in 2006 at a US academic medical center, Schectman et al. found that 65% had not filed an incident report in the past year despite the majority witnessing at least three safety events in that time [21]. While the percentage of physicians failing to file incident reports has remained persistently elevated, some researchers have showed pockets of improvement. Fox et al. implemented a multidimensional intervention that educated resident physicians regarding patient safety, integrated it into their daily work, and addressed the barriers to incident reporting when serious harm events went down and reporting went up [22].

Gaining a better understanding of the barriers to reporting is the first step in effectively addressing them. In our survey, most non-reporting residents indicated that they were not familiar with the procedure and the different criteria for reporting. This was similar to Kaldjian et al.'s survey, which found that approximately only half of the physicians in teaching hospitals knew how to report errors and only 40% were aware of the types of errors to report [9]. Some efforts have been made to address these knowledge deficits in physicians-in-training in Japan. However, much work is still required to achieve a large-scale impact [23]. A clear national set of criteria defining which safety incidents should be reported must be developed and disseminated, similar to what the Joint Commission has done with sentinel events in the US [24]. Interestingly, the Japanese government's Comprehensive Measures for the Promotion of Medical Safety does not clarify the criteria for reporting incidents [25]. Additionally, individual institutions need to ensure that all medical personnel are familiar with how the reporting system works at their facility and find ways to integrate patient safety into daily work. Additionally, non-reporters more commonly indicated that witnessing senior physicians not reporting a safety incident was a barrier to their own reporting. This highlights the need to effectively model this behavior and develop "group norms" of recognizing that it is the responsibility of every person on the medical team to improve the systems of care and support patient safety [26].

Unsurprisingly, the respondents who reported an incident in the previous year were more familiar with the process. However, they more commonly noted the time burden associated with reporting as well as the lack of feedback and noticeable improvement as a result of reporting incidents as barriers. Krouss et al., in their study in 2019, also found these issues to be commonly reported barriers among physician trainees in the US [27]. Similarly, prior reports have shown that people seek, yet rarely receive, feedback on reported incidents [28, 29], despite the fact that feedback was shown to help increase safety awareness, improvement, and motivation [30, 31]. For users to continue to report future safety incidents, efficient and transparent systems are important, so that their value is readily apparent.

Efforts to support increased resident physician reporting of safety incidents are sorely required to address each of the identified barriers. Educating residents on the process and criteria of reporting as well as establishing a group norm will broaden involvement and encourage previously non-reporting residents to engage in this process. Moreover, making incident reporting systems easier and quicker to use, and providing feedback on the impact and changes made as a result of the report, will promote continued reporting of future patient safety incidents. Resident physicians should be encouraged to participate in this aspect of the patient safety movement and stay engaged. Although there are challenges, the potential to bring about change is profound. For example, for countries such as Japan, where there is a national incident reporting system [4], there is potential to analyze and address the themes identified locally and those that may impact patients nationally.

## Limitations

This study had several limitations. First, it was based on a questionnaire survey and did not measure actual report submission behavior. As a result, reporting bias may have influenced the results. Second, the reporting standards for incident reports and the content of safety training varied among hospitals as the criteria of incident reporting are not clearly defined in Japan. Finally, the statistics were based mainly on the reporting and non-reporting groups from previous reports. Therefore, the barriers may change if the percentage of future reporting residents increases. Furthermore, the factors of incident reporting are complex and should not be applied in a general way, as they vary greatly based on the educational system and cultural background of each country.

Despite these limitations, the data for incident reporting revealed by this study is very important. More than half of the residents in Japan did not have reporting experience, despite the fact that guidance for residents requires them to experience incident reporting during residency at a minimum level [10]. Therefore, this should be quickly remedied. This study was a nationwide survey that examined the barriers to incident reporting among residents according to their experience, which we believe will serve as a cornerstone to provide specific strategies to promote safety activities in both reporting and non-reporting groups.

## Conclusion

This study revealed that the barriers for incident reporting among residents were different and greatly dependent on prior experiences with incident reporting. The non-reporting group should be educated regarding reporting procedures and criteria and the reporting group should understand the measures to reduce their hinderances in and the significance of reporting. In the future, respective measures should be taken according to the presence or absence of incident reporting experience to promote the activation of the nationwide reporting campaign.

## Acknowledgments

In preparing this paper, we have relied on the works of Dr. Sanjay Saint, professors at the University of Michigan, and other leading general physicians and outstanding researchers in healthcare quality and safety, in the US, for numerous insights and suggestions. We also thank the team members of the specified non-profit corporation Japan Institute for Advancement of Medical Education Program (JAMEP) for their data collecting support.

## Author Contributions

**Conceptualization:** Takashi Watari, Jeffrey M. Rohde, Ashwin Gupta, Yasuharu Tokuda.

**Data curation:** Masaru Kurihara, Takashi Watari, Yasuharu Tokuda, Yoshimasa Nagao.

**Formal analysis:** Masaru Kurihara, Takashi Watari.

**Funding acquisition:** Masaru Kurihara, Takashi Watari, Yasuharu Tokuda, Yoshimasa Nagao.

**Investigation:** Masaru Kurihara, Takashi Watari, Jeffrey M. Rohde, Yoshimasa Nagao.

**Methodology:** Masaru Kurihara, Takashi Watari, Yasuharu Tokuda.

**Project administration:** Masaru Kurihara, Takashi Watari.

**Resources:** Masaru Kurihara, Takashi Watari.

**Software:** Masaru Kurihara, Takashi Watari.

**Supervision:** Masaru Kurihara, Takashi Watari, Jeffrey M. Rohde, Ashwin Gupta, Yasuharu Tokuda, Yoshimasa Nagao.

**Validation:** Masaru Kurihara, Takashi Watari, Jeffrey M. Rohde, Ashwin Gupta.

**Visualization:** Masaru Kurihara, Takashi Watari, Ashwin Gupta.

**Writing – original draft:** Masaru Kurihara, Takashi Watari, Ashwin Gupta.

**Writing – review & editing:** Masaru Kurihara, Takashi Watari, Jeffrey M. Rohde, Ashwin Gupta, Yasuharu Tokuda, Yoshimasa Nagao.

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
