## [Decision Letter · Decision Letter 0]

30 Aug 2022

PONE-D-22-21250Nationwide survey on Japanese residents’ experience with and barriers to incident reportingPLOS ONE

Dear Dr. Watari, Thank you for submitting your manuscript to PLOS ONE. After careful consideration, we feel that it has merit but does not fully meet PLOS ONE’s publication criteria as it currently stands. Therefore, we invite you to submit a revised version of the manuscript that addresses the points raised during the review process.

We look forward to receiving your revised manuscript.

Kind regards,

Soham Bandyopadhyay

Academic Editor

PLOS ONE

Journal Requirements:

"We also thank the team members of the specified non-profit corporation Japan Institute for Advancement of Medical Education Programme (JAMEP) for their

cooperation."

"This work was supported by the national academic research grant funds [JSPS KA-KENHI: 17K15745, 20H03913]. The sponsor of the study had no role in the study design, data collection, analysis, or preparation of the manuscript."

Reviewers' comments:

Reviewer's Responses to Questions

**Comments to the Author**

1. Is the manuscript technically sound, and do the data support the conclusions?

Reviewer #1: Yes

Reviewer #2: Partly

2. Has the statistical analysis been performed appropriately and rigorously? 

Reviewer #1: Yes

Reviewer #2: No

3. Have the authors made all data underlying the findings in their manuscript fully available?

Reviewer #1: Yes

Reviewer #2: Yes

4. Is the manuscript presented in an intelligible fashion and written in standard English?

Reviewer #1: Yes

Reviewer #2: Yes

5. Review Comments to the Author

Reviewer #1: the manuscript is written in standard English. Statistical analysis also have been performed appropriately and rigorously and the results of the analysis were coherent with the discussion and conclusion part of the manuscript.

Reviewer #2: The current study can be considered the improvement key of safety incident reporting systems, not ‎just in Japan, but also in other countries, taking into consideration the different healthcare systems ‎and the available resources. Inaddition, it is a nationwide survey that played an integral role in data ‎representation of the population in Japan, so applaud the initiative. ‎

There are a few points which may improve the present manuscript are enlisted below:‎

‎1. Standardization in presenting data within the context where it is advisable to write the frequency ‎then the percentage in brackets as in lines 32, 33, 34 and throughout the manuscript.‎

‎2. The manuscript needs a slight English proofreading for grammar, as the statement in the abstract ‎line 39 - 41 and line 58 use another conjunctive to clarify the content message. ‎

‎3. Citation: the reference number should be added then the dot. as statement [1]. instead of ‎statement. [1] with standardization throughout the manuscript.‎

‎4. References list need to be reviewed for Vancouver style, specifically the websites need to add the ‎access date.‎

‎5. In the study design line 79 the author has mentioned the questionnaire was used from previous ‎study "We used a validated

‎ questionnaire from a previous study," but then in line 101 they stated it was developed by the two ‎authors, thus contradictory statements, while reviewing the previous cited studies [13] [15] and [16] ‎the authors TW was not one of the authors.‎

‎6. Human Consideration: it is better to clarify how the consent was obtained from participants ‎‎(verbal/written)‎

‎7. Abbreviations: in line 101 it is better to specify the authors' full names and not just their initials.‎

‎8. Results: the data presentation in Table 1 is missing the female data raw. also data should be ‎presented more clearly as it is advisable to merge the title cells of the title characteristics table , and to ‎standardize data presentation as n= then (%) in the tables and within the context of the manuscript.‎

‎9. there were no median neither mean analysis of the continuous variables despite what the authors ‎had mentioned in the Statistical analyses line 132-133.‎

‎10. Fig 2 should be uploaded with a better resolution to be able to visualize statistical parameters more ‎clearly.‎

‎11. Author contributions are missing, as per the publication guidelines for PLOS ONE.‎

6. PLOS authors have the option to publish the peer review history of their article (what does this mean?). If published, this will include your full peer review and any attached files.

Reviewer #1: No

Reviewer #2: **Yes: **Alaa Dayekh

---

## [Author Response · Author response to Decision Letter 0]

14 Oct 2022

October 14, 2022

Dear Editor:

Thank you for inviting us to submit a revised draft of our manuscript titled “Nationwide survey on Japanese residents’ experience with and barriers to incident reporting.” We appreciate the time and effort you and each of the reviewers have dedicated to providing insightful feedback on ways to strengthen our paper. We have incorporated changes that reflect the detailed suggestions you have graciously provided. We hope that our edits and the responses we provide below satisfactorily address all the issues and concerns you and the reviewers have noted.

To facilitate your review of our revisions, the following is a point-by-point response to the questions and comments delivered in your letter.

Thank you for your consideration. I look forward to hearing from you.

Sincerely,

Takashi Watari, M.D, MHQS, MCTM, Ph.D.

Reviewer 2: 

Comment 1: Standardization in presenting data within the context where it is advisable to write the frequency ‎then the percentage in brackets as in lines 32, 33, 34 and throughout the manuscript.‎

Response 1: Thank you for your observation. We agree with your comments and have revised the abstract as follows: 

Of 5810 respondents, the vast majority indicated they had encountered at least one safety incident in the past year (n=4449, 76.5%). However, only 2724 (46.9%) had submitted an incident report. Under-reporting (more safety incidents compared to the number of reports) was evident in 1523 (26.2%) respondents. The most frequently mentioned barrier to reporting an incident was the time required to file the report (n=2622, 45.1%).

Comment 2: The manuscript needs a slight English proofreading for grammar, as the statement in the abstract ‎line 39 - 41 and line 58 use another conjunctive to clarify the content message. ‎

Response 2: Thank you for your helpful suggestion. We have revised them as follows: 

Line 39-41: Our study revealed that resident physicians in Japan commonly encounter patient safety incidents but under-report them.

Line 58: Therefore, they are often the first to encounter safety incidents.

Comment 3: Citation: the reference number should be added then the dot. as statement [1]. instead of ‎statement. [1] with standardization throughout the manuscript.‎

Response 3: Thank you for your suggestions. I have placed the reference numbers before the punctuation and have ensured consistency throughout the manuscript.

Comment 4: References list need to be reviewed for Vancouver style, specifically the websites need to add the ‎access date.‎

Response 4: Thank you for your valuable suggestion. Accordingly, we have revised the reference style especially for the website references. 

Comment 5: In the study design line 79 the author has mentioned the questionnaire was used from previous ‎study "We used a validated

‎ questionnaire from a previous study," but then in line 101 they stated it was developed by the two ‎authors, thus contradictory statements, while reviewing the previous cited studies [13] [15] and [16] ‎the authors TW was not one of the authors.

Response 5: We appreciate your helpful suggestions. It means that GM-ITE has been conducted and used in research. Therefore, we have revised the manuscript on page 5 as follows: 

“Both the original GM-ITE and the abovementioned questionnaire have been used in prior research studies.” (lines 85-86)

Comment 6: Human Consideration: it is better to clarify how the consent was obtained from participants ‎‎(verbal/written).‎

Response 6: Thank you for the valuable feedback. The consent in this study was obtained from the participants ‎‎in the written form. Therefore, we revised the text on page 6 as follows: 

“Informed consent was obtained from all participants in the written form.” (lines 87-88)

Comment 7: Abbreviations: in line 101 it is better to specify the authors' full names and not just their initials

Response 7: Thank you for the valuable feedback. We revised the text on page 6 as follows:

“The questionnaire was developed following consensus among two investigators (Masaru Kurihara and Takashi Watari) based on known incident reporting challenges in Japan (e.g., under-reporting and lack of knowledge on patient safety) [13].” (lines 100-102)

Comment 8: Results: the data presentation in Table 1 is missing the female data raw. also data should be ‎presented more clearly as it is advisable to merge the title cells of the title characteristics table, and to ‎standardize data presentation as n= then (%) in the tables and within the context of the manuscript.‎

Response 8: We appreciate your helpful and valuable suggestions. We revised Table 1 as seen in pages 8-9. I have merged the required cells and added the data for female respondents. Moreover, the data has been presented as frequency and percentages in a consistent format.

Comment 9: there were no median neither mean analysis of the continuous variables despite what the authors ‎had mentioned in the Statistical analyses line 132-133.‎

Response 9: Thank you for your keen observation. This sentence has been removed.

Comment 10: Fig 2 should be uploaded with a better resolution to be able to visualize statistical parameters more ‎clearly.‎

Response 10: Thank you for your valuable suggestion. We have ensured that Fig 2 is clearer.

Comment 11: Author contributions are missing, as per the publication guidelines for PLOS ONE.

Response 11: Thank you for your valuable suggestion. Accordingly, we have added the information on author contributions at the end of the manuscript.

“Author contributions

M.K., T.W., Y.T, and Y.N. designed the study, main conceptual ideas, and proof outline. M.K, Y.T, and Y.N collected the data. T.W., J.R. and A.G. aided in interpreting the results and prepared the manuscript. T.W. and Y.T supervised the project. M.K., and T.W. wrote the manuscript with support from J.R. and A.G. All authors discussed the results and commented on the manuscript.

”

---

## [Editor Report · Decision Letter 1]

21 Nov 2022

Nationwide survey on Japanese residents’ experience with and barriers to incident reporting

PONE-D-22-21250R1

Dear Dr. Watari,

We’re pleased to inform you that your manuscript has been judged scientifically suitable for publication and will be formally accepted for publication once it meets all outstanding technical requirements.

Kind regards,

Soham Bandyopadhyay

Academic Editor

PLOS ONE
---

## [Editor Report · Acceptance letter]

24 Nov 2022

PONE-D-22-21250R1 

Nationwide survey on Japanese residents’ experience with and barriers to incident reporting 

Dear Dr. Watari:

I'm pleased to inform you that your manuscript has been deemed suitable for publication in PLOS ONE. Congratulations! Your manuscript is now with our production department. 

Kind regards, 

on behalf of

Dr. Soham Bandyopadhyay 

Academic Editor

PLOS ONE